# Management of Cutaneous Head and Neck Squamous and Basal Cell Carcinomas for Immunocompromised Patients

**DOI:** 10.3390/cancers15133348

**Published:** 2023-06-26

**Authors:** Krishna K. Bommakanti, Nikitha Kosaraju, Kenric Tam, Wanxing Chai-Ho, Maie St. John

**Affiliations:** 1Department of Head and Neck Surgery, David Geffen School of Medicine, University of California, Los Angeles, Los Angeles, CA 90095-1624, USA; 2UCLA Head and Neck Cancer Program (HNCP), David Geffen School of Medicine, University of California, Los Angeles, Los Angeles, CA 90095-1624, USA; 3David Geffen School of Medicine, University of California, Los Angeles, Los Angeles, CA 90095-1624, USA; 4Department of Medicine, Division of Hematology/Oncology, David Geffen School of Medicine, University of California, Los Angeles, Los Angeles, CA 90095-1624, USA

**Keywords:** cutaneous squamous cell carcinoma, basal cell carcinoma, non-melanoma skin cancer, immunotherapy

## Abstract

**Simple Summary:**

Non-melanoma skin cancer affects a significant portion of the population in the United States, with over one million cases diagnosed each year. Skin cancers in the head and neck are considered high risk for locoregional spread and recurrence, requiring close monitoring and multidisciplinary management by head and neck surgeons, oncologists, radiation oncologists, and many others. In this study, we performed an extensive literature review to summarize current knowledge regarding the etiology, disease course, and management of head and neck skin cancers in immunocompromised patients. We draw attention to the role of newly developed immunotherapies being used in this subset of patients.

**Abstract:**

The incidence of non-melanoma skin cancer (NMSC) continues to rise, and more than one million cases are diagnosed in the United States each year. The increase in prevalence has been attributed to increased lifespan and improvements in survival for conditions that increase the risk of these malignancies. Patients who are immunocompromised have a higher risk of developing NMSC compared to the general population. In immunosuppressed patients, a combination of prevention, frequent surveillance, and early intervention are necessary to reduce morbidity and mortality. In this review, we collate and summarize current knowledge regarding pathogenesis of head and neck cutaneous SCC and BCC within immunocompromised patients, examine the potential role of the immune response in disease progression, and detail the role of novel immunotherapies in this subset of patients.

## 1. Introduction

Non-melanoma skin cancers (NMSCs) comprise one-third of all malignancies in the United States [1,2]. The incidences of basal cell carcinoma (BCC) and cutaneous squamous cell carcinoma (cuSCC) worldwide have risen by 3–10% and 4–14%, respectively, each year [1,3]. The incidence of cuSCC has been reported to range from 129–208 cases per 100,000 persons and the incidence of BCC ranges from 293–360 cases per 100,000 persons [4].

Patients who are immunocompromised have an increased risk of developing NMSCs compared to the general population [5,6]. Immunocompromised patients are defined by the ID society as a group of individuals with acquired or inherited immune deficits that affect multiple parts of the immune system. Immunocompromised patients lack some or all of the immune defense mechanisms required to prevent tumorigenesis and are therefore susceptible to many cutaneous malignancies.

Organ transplant recipients (OTRs) are a subset of immunocompromised patients who are particularly susceptible to NMSCs. These patients are 65 to 250 times more likely to develop cuSCC and 10 to 16 times more likely to develop BCC [2,6]. Although the overwhelming majority of these cancers can be managed with surgery alone, immunocompromised patients are more likely to present with high-risk pathologic features, aggressive phenotypes, and poorer outcomes [5,7,8,9,10,11]. Understanding the disease progression of BCC and cuSCC in the immunocompromised population is particularly important given the significant alterations in prognosis and response to treatment within this group. Locoregional recurrence rates after surgery alone in immunocompromised patients range between 13% and 48%; the rate of distant metastases ranges between 7% and 19%. These rates are in stark contrast to those in the general population, where locoregional and distant metastases are very rare occurrences [5,6,10,12,13,14].

Multidisciplinary efforts involving prevention, surveillance, and early intervention remain essential for decreasing the morbidity and mortality of NMSC in immunocompromised patients. Figure 1 provides a summary of the current treatment paradigm for cuSCC and BCC in immunocompromised and immunocompetent patients. In this review, we present an overview of the pathophysiology and management of cuSCC and BCC in the head and neck with specific emphasis on the unique range of treatment approaches and advances in treatments for cuSCC and BCC in immunocompromised patients.

## 2. Pathogenesis of NMSCs

### 2.1. Pathogeneisis of CuSCC

CuSCCs often start as non-cancerous actinic keratoses (AKs). While only approximately 10% of AKs evolve into cuSCC, a majority of cuSCCs arise from AKs, with one report identifying that 72% of cuSCC cases developed from an AK [15,16]. The main risk factor for AKs is UVB exposure [17,18]. Excess UVB exposure can lead to UVB “signature” mutations in DNA, which consist of C to T and CC to TT transition mutations [17,19]. These transition mutations can lead to inactivation of p53, a tumor suppressor protein that is often mutated in AKs [17]. While UVB damage creates the initial changes necessary for AK formation, cuSCC develops as UVB-exposed keratinocytes that undergo clonal expansion through a series of additional mutations in oncogenes and anti-oncogenes. A number of additional genetic mutations have also been implicated in the pathogenesis of cuSCC, including *BCL2*, *RAS*, and *p53*. In 2019, Zhao et al. demonstrated that melanoma-associated antigen gene A12 (*MAGEA12*)’s product is significantly increased in cuSCC, as it downregulates p21, a protein that facilitates cell growth arrest and therefore DNA repair processes [20]. Another protein responsible for recognition of UVB-induced DNA damage and nucleotide excision repair is XPC [21]. The *XPC* gene has been found to be inactivated or lost in cuSCC patients [21]. Changes in the intracellular signal transduction pathways such as EGFR and COX can also lead to the development of invasive cuSCC [17]. Preliminary work on the contributions of Hedgehog signaling in pathogenesis of cuSCC has shown that in a subset of cuSCC, Hedgehog signaling can antagonize cuSCC initiation, proliferation, and migration [22,23,24].

### 2.2. Pathogenesis of cuSCC in Immunocompromised Patients

Several studies suggest that the intensity and duration of immunosuppression and the development of cuSCC are directly correlated. In a systematic review by Jiyad et al. OTRs treated with azathioprine, a common anti-proliferative and immunosuppressive agent, were found to have a significant increased risk of cuSCC (1.56, 95% confidence interval (CI) 1.11–2.18) but not BCC [25]. Renal transplant patients receiving a three-drug immunosuppressive regimen (azathioprine, cyclosporine, and prednisolone) had a 2.8 times greater risk of developing cuSCC compared to OTRs receiving only azathioprine and prednisolone [26]. Azathioprine is thought to increase the risk of cutaneous malignancies through UVA photosensitization, resulting in DNA damage through oxidative stress [17,27,28,29]. Cyclosporine and tacrolimus, both immunosuppressants, have been shown to have synergistic effects with UVA and UVB, increasing tumorigenesis by reducing DNA repair, increasing angiogenesis and inflammation, and preventing p53-dependent cell senescence [17,27,30,31].

Not all immunosuppressant medications carry the same risk of NMSC development. mTOR inhibitors (e.g., sirolimus and everolimus) and mycophenolate mofetil (MMF) have a reduced incidence of NMSC compared to calcineurin inhibitors [27,32,33,34]. In 2016, Coghill et al. performed a retrospective case–control study to compare the rate of developing cuSCCs with the use of calcineurin agents or mycophenolic acid agents such as MMF in renal and cardiac transplant patients [35]. They demonstrated that users of the older anti-metabolite azathioprine were more than twice as likely to develop cuSCCs within two years of having undergone a transplant [35]. mTOR inhibitors have similarly been shown to carry a relatively reduced risk of NMSC development. In a systematic review by Knoll et al., sirolimus was associated with a lower risk of NMSC (and malignancy overall) in kidney transplant patients [34]. Across all studies included in the meta-analysis, the authors found a 40% reduction in the risk of malignancy and a 56% reduction in risk of NMSC for those randomized to sirolimus [34].

Immunocompromised patients are particularly susceptible to fungal infections and are therefore oftentimes treated with voriconazole, a broad-spectrum anti-fungal. Voriconazole is thought to lead to cuSCC development through increased absorption of UVA and UVB by voriconazole’s primary metabolite, voriconazole N-oxide [17,27,36]. Along with fungal infections, immunosuppression can lead to higher risk of acquiring human papillomavirus (HPV) infections. In particular, HPV of the beta genus (betaPV) has been implicated in cuSCC development with betaPV also found to be synergistic with UV damage [27].

### 2.3. Pathogenesis of BCC

The pathogenesis of BCC is primarily driven by the Sonic Hedgehog (SHH) pathway and *TP53*, a tumor suppressor gene. Mutations of the Hedgehog pathway, specifically in the patched 1 (*PTCH1*) gene, can be a result of UV exposure and oxidative stress, leading to hyperactivation of the Hedgehog pathway, resulting in increased gene transcription [37,38,39]. BCC tumors have been found to have high rates of *TP53* mutations, which lead to a loss of genomic integrity since the *TP53* gene encodes the P53 protein, a regulatory protein involved in DNA repair [37,38,39]. A recent review article by Hoashi et al. on molecular mechanisms and targeted therapies of BCC emphasizes the role of the melanocortin 1 receptor, MC1R, stating that several studies have reported that the risk of BCC is associated with MC1R mutations [39,40,41,42]. It is known that melanocytes with MC1R loss of function mutations have a decreased capacity to repair DNA damage caused by UV light, but this has yet to be shown with keratinocytes [39,40]. Other molecular factors that have been implicated in the pathogenesis of BCC include Wnt signaling, *SOX2*, and geminin (GMNN) [43,44,45].

### 2.4. Pathogenesis BCC in Immunocompromised Patients

Immunocompromised patients are much more likely to develop cuSCC than BCC due to the differences in pathogenesis and the synergistic role many immunosuppressants have with UV damage implicated in cuSCC development. However, immunocompromised patients do still have an increased risk of developing BCC when compared to the immunocompetent population [2,6]. Immunosuppression results in an inability to repair DNA damage and detect malignant cells, allowing not only cuSCC to develop, but BCC as well [46]. It has been well established that immunosuppressants are linked to development of NMSC; however, a limitation of these previous studies is the grouping of cuSCC and BCC together under the umbrella term NMSC. Due to this, the mechanisms behind the increased risk for BCC in immunocompromised patients are not well elucidated. Conversely, the mechanisms for cuSCC are given that the cuSCC risk in immunocompromised patients is exponential while for BCC it is linear [46].

## 3. Role of Immune Evasion in the Progression of cuSCC and BCC

As detailed above, immunocompromised individuals are at significantly higher risk for developing NMSC. Although the pathogenesis of cuSCC and BCC is well understood, the role of the tumor immune microenvironment (TIME) in the progression of NMSC is still under study [47,48,49,50]. TIME is defined as the composition of immune infiltrate and consists of cancer cells, extracellular matrix, and stromal cells. Broadly, the adaptive immune system interacts with intracellular proteins through the human leukocyte antigen (HLA) pathway [47,51]. Expressed proteins are presented to HLA cell-surface proteins and mutations in these proteins can modify the immune system’s ability to recognize and respond to them [51,52]. Tumor surveillance is performed by the host immune system and the majority of tumor cells are detected and eliminated. Over time, less immunogenic tumor cells evade these immune mechanisms and promote tumor formation [47]. In immunocompromised patients with NMSC, the increased infiltration of specific inflammatory cells happens during skin carcinogenesis and is associated with more aggressive disease and metastasis [47].

### 3.1. Role of TIME in cuSCC

TIME can aid immune evasion in cuSCC through several mechanisms. One important mechanism by which tumors evade detection is through the selection of poorly immunogenic tumor cells. Amor et al. and others have identified the ability of the TIME in cuSCC to suppress the host’s activated immune response and delay the initiation of response via alterations in T cell proliferation and the release of several inflammatory cytokines [12,47,48,53,54]. Two pathways of immune suppression—cytotoxic T-lymphocyte-associated protein 4 (CTLA-4) and programmed cell-death protein-1 (PD-1)—have been implicated in the development and progression of cuSCC and are the most well-studied examples of T cell immune checkpoint molecules that are used to evade host immunity [47,55,56,57,58,59,60,61]. Given the critical role of CTLA-4 and PD-1 in the tumorigenesis of cuSCC, several clinical trials are looking at a blockade of one or both of these pathways via monoclonal antibodies to the receptors (Table 1).

CTLA-4 is a transmembrane molecule that downregulates T cell activity [47]. Loser et al. used mouse models of UV-induced tumors such as NMSCs to identify the key role of CTLA-4 in the development of NMSCs as well as its involvement in generating anti-tumor memory responses [60,61]. The in vivo and in vitro CTLA-4 blockade decreased the activity of UV-induced Tregs, thus suggesting that the inhibition of this pathway is protective against tumor growth in the TIME [47,61].

T cell activation is also regulated by PD-1 and its ligands, PD-L1 and PD-L2, which are highly expressed in cuSCC. Belai et al. demonstrated in a murine model that blocking PD-1 resulted in a strong anti-tumoral response that was characterized by an increase in activated T cells and a reduction of TGF- β, which acts as an immunosuppressive cytokine [47,62].

### 3.2. Role of TIME in BCC

Similar to its role in cuSCC, the TIME in BCC can promote immune evasion and tumor progression through several mechanisms, including overexpression of tumor-infiltrating T cells and a high rate of genetic mutational burden [63,64]. As with cuSCC, the CTLA-4 and PD-1 receptors are considered key pathways in allowing for unchecked tumor growth and several of the same monoclonal antibodies are also being studied in BCC (Table 1) [63].

In addition to these pathways, BCCs also demonstrate overactivation of the SHH pathways and concomitant low levels of major histocompatibility complex-1 (MHC-I) expression, which may suppress anti-tumor immunity and allow for immune escape [63]. MHC-1 molecules are expressed on the surface of all nucleated cells and are important for identifying tumor-affected cells and activating the host immune response [65]. Upregulation of the SHH pathway is thought to be tied to the low MHC-1 levels seen in BCC and small molecule inhibitors targeting the SHH pathway are under study (Table 2).

## 4. Surgical Management of cuSCC and BCC in Immunocompromised Patients

### 4.1. SCC

Surgical excision with wide margins remains the standard of care for management of cuSCC [2,5,66,67]. Many studies have demonstrated that immunocompromised patients with cuSCC are more likely to have aggressive disease and that, compared with immunocompetent patients, they have a significantly lower locoregional recurrence-free survival [2,5,68,69,70,71,72,73,74]. Specifically for head and neck cuSCC, immunosuppression is associated with a 2.32 times increased risk of disease-specific death [75]. An immunosuppressed patient diagnosed with cuSCC has 3 times the odds of having a more aggressive course than a non-immunosuppressed patient [2,5,71]. SCCs in immunocompromised patients are therefore treated as high risk regardless of tumor diameter given their propensity for recurrence, locoregional spread, and metastasis [2,76,77].

For immunosuppressed patients with high risk cuSCC, NCCN guidelines recommend wide excision with deep and peripheral margin assessment [2,70,71]. Recommended excision margins are at least 6 mm and in several cases > 1 cm [2,66,70,71]. If Moh’s micrographic surgery is pursued, at least three stages are recommended in immunocompromised patients [2,5,70,71]. Other important recommendations in this patient population include sentinel lymph node (SLN) biopsy (even without clinically apparent LAD) and postsurgical radiation therapy [70,71,78,79]. In the case of lymph node involvement by cuSCC, the preferred treatment is a regional lymph node dissection [70,71,78].

### 4.2. BCC

BCC in immunosuppressed patients is classified as high risk regardless of location, diameter, or other tumor-specific factors [2,3,79,80]. An immunosuppressed patient diagnosed with BCC has 2–3 times the odds of having a more aggressive course than a non-immunosuppressed patient. Recommended excision margins follow the NCCN guidelines for high-risk BCC [2,3,78,81,82,83].

BCC is characterized by subclinical extension beyond the visible tumor and, for this reason, standard excision should include a margin of clinically normal-appearing skin [2,79,80,82,84]. The NCCN guidelines recommend the use of margin assessment via Moh’s surgery or intraoperative frozen sections [2,80,81,85]. If this is not feasible, standard surgical excision is recommended with margins > 4 mm [80,81]. In the treatment of high-risk facial BCC, studies have demonstrated that Moh’s surgery resulted in fewer recurrences compared to standard surgical excision over a 10-year period [3,80,81]. Additional recommendations include the use of adjuvant radiation therapy for high-risk BCC [3,80,81].

## 5. Radiation Therapy in the Management of cuSCC and BCC

Radiation therapy (RT) is seldom used as a primary or single treatment modality in the management of cuSCC. Although it remains an option for primary treatment of cuSCC when surgery is contraindicated, primary RT requires a prolonged treatment course and carries the potential risk of disease recurrence. Per AAD guidelines, adjuvant RT is recommended in patients with high-risk features of cuSCC including perineural invasion, positive margins, or evidence of metastatic disease. The efficacy of adjuvant RT has not been clearly established, though it has been shown in retrospective studies to reduce the likelihood of recurrence and metastasis after surgical excision [49,86,87]. The role of RT in the management of cuSCC is not as clearly understood in the immunocompromised population. Despite the increased risk of locoregional metastases with high-risk cuSCC and a recurrence rate of 33–50% in these patients, many authors report that radiation therapy is not recommended as a first line or single modality treatment option in OTRs since these patients are likely to develop multiple NMSCs [88,89]. The aggressive nature of the disease in this population, however, is more likely to result in high-risk features for which adjuvant radiation is strongly recommended.

RT is considered as a primary treatment for local low-risk and high-risk BCCs in non-surgical candidates, and a wide range of radiation techniques and dosing protocols are in use [90]. As in the immunocompetent population, BCC is rarely likely even amongst chronically immunosuppressed patients, and the recommendations for management of BCC are almost identical between the two [89]. For adjuvant BCC treatment, NCCN guidelines recommend RT for tumors with high-risk features, including perineural invasion and positive margins after Moh’s excision or wide local excision. Other high-risk features warranting RT are involvement of surrounding muscle, cartilage, or bone [90].

The benefit of radiation therapy as a sensitizing agent that can be used in conjunction with immunotherapy is under investigation for its use in cuSCC and BCC. Radiation therapy has been shown to modulate the TIME by increasing the release of cytokines and stimulating the proliferation of immune cells, which makes these tumors more susceptible to targeted immunomodulators [91]. The synergistic effect of stereotactic radiotherapy in conjunction with PD-1 inhibitors has been proven in Merkel cell carcinoma but is still under study as a treatment modality in patients with cuSCC and BCC [91].

## 6. Current Systemic Therapies and Ongoing Clinical Trials

Most patients with early-stage localized NMSCs can be successfully treated with local therapy, such as surgery or radiation therapy. In recurrent or metastatic disease, platinum-based chemotherapy or targeted therapy against epidermal growth factor receptor (EGFR) with cetuximab can be utilized. Based on the encouraging efficacy and favorable side effect profile, immune checkpoint inhibitors (immunotherapy) have become the first-line systemic treatment options for non-immunocompromised patients with metastatic disease or those with locally advanced disease who are unable to undergo surgery or radiation.

As of March 2023, using clinicaltrials.gov, the search terms “cutaneous squamous cell carcinoma” and “immunotherapy” identified 44 studies. When filtered to only include those administering drug(s), 37 studies remained. The national clinical trial (NCT) number, clinical trial title, clinical trial status, agent(s) used in the study, and phase of the study for cuSCC are shown in Table 1. With the search terms “cutaneous basal cell carcinoma” and “immunotherapy”, five studies were identified. After removal of clinical trials that do not involve immunotherapy agent(s), there were four studies. The national clinical trial (NCT) number, clinical trial title, clinical trial status, agent(s) used in the study, and phase of the study for BCC are shown in Table 2. Table 3 identifies the subset of clinical trials that include immunocompromised patients. The same search terms were used with the addition of “immunosuppression” to identify clinical trials which include these patients.

### 6.1. Systemic Therapy for cuSCC

As discussed earlier, immune evasion and modification of the TIME are major mechanisms by which SCC grows and develops. Targeting various pathways within the TIME to slow or halt tumor progression is a major area of research. Table 1 provides a full list of ongoing clinical trials involving systemic therapies for cuSCC, a subset of which will be discussed below. The relationship between the PD-1 receptor and its ligand, PDL-1, has shown particular promise in promoting the death of cancerous cells in cuSCC. Cemiplimab is a monoclonal antibody that acts as a PD-1 receptor inhibitor. PD-1 inhibitors prevent the interaction of the programmed death-ligand 1 (PD-L1) with its receptor, PD-1 [92]. When this ligand interacts with its receptor, the immune system is suppressed, allowing cancerous cells to proliferate [86,92]. The FDA approval of cemiplimab was based on reports from a phase 1/2 study in patients with locally advanced or metastatic cuSCC [93]. In the phase 1 study, 50% of patients had an objective response and in the phase 2 study, objective responses were observed in 47% of patients with metastatic disease and in 60% of patients with unresectable, locally advanced disease [93,94]. Treatment was well tolerated, with treatment discontinuation rate of 5% due to adverse events. The long-term follow-up data presented at the American Society of Clinical Oncology meeting in 2020 showed that in responding patients, the estimated proportion of patients with ongoing response at 24 months was 69.4% [95]. Pembrolizumab is another PD-1 inhibitor that was extensively studied for cuSCC. The overall response rate was 50% for locally advanced, unresectable patients, with complete response rate of 17% [96,97]. In those with locally advanced recurrent or metastatic disease, the objective response rate was 35%, with complete response rate of 10%. Among the patients with disease response, 68% had disease responses lasting 12 months or longer [98].

### 6.2. Systemic Therapy for BCC

The molecular pathways that result in the tumorigenesis of cuSCC are also implicated in BCC. Cemiplimab, which is discussed above, is a PD-1 receptor antibody approved for use in locally advanced or metastatic BCC that is unresponsive to first-line agents. In those patients with locally advanced disease, 26 of 84 patients (31%) were found to have an objective response [99]. Based on this study, cemiplimab was approved for patients with locally advanced or metastatic BCC; however, further data are needed to evaluate the objective response rate in chronically immunosuppressed patients.

Small molecule inhibitors and monoclonal antibodies are also currently used in the management of high-risk and widely metastatic BCC. Vismodegib and sonidegib both target the Hedgehog pathway, which is most commonly involved in BCC. A 2009 phase 1 trial of patients on vismodegib taken at a dose of 150 mg daily had an objective response rate of 58% [90,100,101]. Results, however, should be interpreted with caution as 20% of patients in the trial progressed due to the development of drug resistance. Vismodegib use in conjunction with other therapeutics is currently being studied [90,100,101]. Sonidegib is another small molecule inhibitor approved for use in locally advanced BCC [90,99,102,103,104,105,106,107]. In a trial of nine patients with advanced BCCs who progressed after treatment with vismodegib, patients were treated with sonidegib for a median of 6 weeks and over half had disease progression [90,105].

### 6.3. Systemic Therapy in Immunocompromised Patients

Despite encouraging efficacy, immunotherapy agents were not initially studied in clinical trials for treatment of NMSC in organ transplant recipients (OTRs) or patients with active autoimmune conditioning due to concern about inducing allograft rejection due to exacerbation of autoimmune activity. Few studies have looked specifically at the effects of systemic therapy in chronically immunosuppressed patients. The search for therapeutic agents that provide good efficacy in immunocompromised patients is an ongoing area of active research. Table 3 includes a list of clinical trials of systemic and chemopreventive agents that have been studied in immunocompromised patients with NMSCs. Aside from capecitabine, which has only been studied in cuSCC, and diclofenac sodium, which has only been studied in BCC, the remaining systemic agents are being tested in patients with either condition.

Capecitabine is a prodrug of 5-fluorouracil (5-FU), which inhibits DNA and RNA synthesis by reducing thymidine and uridine triphosphate production. A case study by Endrizzi et al. found that capecitabine halted the rate of cuSCC tumor development over a 12-month period in ten kidney and liver transplant patients [108,109].

Diclofenac sodium is a non-steroidal anti-inflammatory agent that inhibits cyclooxygenase (COX) 2 [109]. It is FDA approved for the treatment of actinic keratoses and has demonstrated efficacy in immunosuppressed patients. A randomized control trial from 2012 demonstrated that in OTRs, twice daily application of diclofenac 3% gel to AKs on the face, hands, and balding scalp for 16 weeks showed complete clearance of AKs in 41% of patients [109,110].

Several systemic therapeutic agents are currently under study for their role in treating cuSCC and BCC in immunocompetent and immunocompromised patients. RP1 is an oncolytic virus (herpes simplex virus-1 (HSV-1)) that expresses a fusogenic glycoprotein (GALV-GP R-) and granulocyte macrophage colony-stimulating factor (GM-CSF) [111]. A phase 1b/2, multicenter, open-label study (NCT04349436) is currently ongoing, evaluating efficacy and safety of RP1 intratumoral injection for the treatment of locally advanced or metastatic cutaneous malignancies (to skin, soft tissue, or lymph nodes) in up to 65 evaluable OTR patients [111].

A recent clinical trial studied the effects of nicotinamide as a chemopreventive agent in NMSCs in organ transplant recipients. Nicotinamide is the amide form of vitamin B3 that prevents UV radiation-induced immunosuppression through its involvement in DNA damage repair [112]. Nicotinamide has been shown to reduce the development of actinic keratoses and associated cancers in the general population but had not until recently been well studied in the organ transplant population [112,113]. Although AKs are the precursor lesion for cuSCC, its use has been studied in both cuSCC and BCC. In 2016, a small case–control study initially suggested that 88% of OTRs treated with nicotinamide had a decrease in size of actinic keratosis while, among the control group, 91% showed an increase in lesion size [114]. A recent phase 3 randomized control trial that was complete in 2023 studied the effect of daily oral nicotinamide in organ transplant patients with a history of at least two NMSCs [112]. One hundred and fifty-eight patients were randomly assigned to nicotinamide or placebo and received 500 mg twice daily for 12 months. The trial was initially set to last five years but was stopped early because of poor recruitment. At the time of conclusion, there were no significant differences observed in SCC counts, BCC counts, actinic keratosis counts, or quality-of-life scores [112].

Additional topical or adjunct therapies have shown to be effective in immunocompromised patients with either cuSCC or BCC. Photodynamic therapy (PDT) in conjunction with topical methyl 5-aminolaevulinate (MAL) cream has also been studied in OTRs [115,116]. Initially, PDT was studied primarily in patients with superficial BCC; however, recent studies have demonstrated that PDT can also be effectively used in the management of cuSCC in situ or its precursor lesion, AK [117]. MAL is a photosensitizing agent used in conjunction with PDT to generate reactive oxygen species that selectively destroy tumor cells [115]. MAL has gained popularity over other photosensitizing agents because of its ability to selectively accumulate and destroy tumor cells through its activation of protoporphyrin IX, a photosensitizing agent [115]. Results from a randomized clinical trial comparing the use of MAL and PDT to topical 5% fluorouracil (5-FU) for the treatment of premalignant lesions in OTRs demonstrated greater effectiveness in the resolution of lesions with use of PDT at one, three, and six months after treatment [115]. Results from a multicenter randomized controlled trial studying the use of PDT with MAL in OTRs with NMSC are pending [116]. Acitretin is a vitamin A derivative that activates retinoid receptors and regulates several cellular functions including differentiation, maturation, and apoptosis [109]. In a randomized controlled trial of renal transplant patients taking 30 mg/day of acitretin, the authors found a statistically significant decrease in the number of cuSCCs seen in patients at the 6- and 12-month marks [109,118].

## 7. Discussion

Head and neck NMSC in patients with immunosuppression is associated with poor outcomes and an aggressive course when compared with NMSC in the immunocompetent population. While surgical and ablative methods are the primary form of treatment for BCC and cuSCC, the options for managing high-risk and widely metastatic NMSC in both immunocompetent and immunocompromised individuals have evolved to include a wide array of treatment options. Treatment modalities include a combination of surgical resection, topical therapy and chemoprevention, radiation therapy and other field treatments, targeted molecular therapies, and immunotherapy. The appropriate choice of therapy requires an understanding of tumor-specific and patient risk factors.

For the treatment of BCCs, the European consensus-based interdisciplinary guidelines recommend complete surgical excision as the first-line treatment for all BCCs [119]. They also recommend the use of topical or photodynamic therapy for low-risk superficial BCC [119]. For high-risk BCCs, a multidisciplinary approach, including the use of systemic and targeted molecular therapies, is required. In particular, the small molecule inhibitors vismodegib and sonidegib that target the SHH pathway have been approved for use in patients with recurrent or metastatic BCC [120,121]. Guidelines for the management of cuSCC similarly involve a combination of surgical excision and topical therapies [3]. Surgical excision via MMS or wide local excision is considered the most effective treatment for cuSCC [3]. In cases where surgery is not an option, radiation, photodynamic therapy, or topical therapy can be considered as alternatives [3]. The management of patients with metastatic cuSCC involves the use of chemotherapeutic agents or radiation, either alone or in conjunction [3]. For high-risk or locally advanced NMSCs with an underlying immunocompromising condition, a multidisciplinary approach including systemic and targeted molecular therapies should also be considered alongside definitive surgery as this has demonstrated improved outcomes. For immunocompromised patients with metastatic NMSCs, referral to a high-volume treatment center for consideration of immunotherapy or clinical trial enrollment should be considered.

Treatment of NMSCs in immunocompromised patients has similarly evolved to include a multidisciplinary approach consisting of surgery, radiation therapy, chemotherapy or immunotherapy, and other targeted therapies. In aggressive cases that have not responded to surgery or immunotherapy, checkpoint inhibitors have demonstrated great potential in immunocompetent patients. Recently, there have been emerging data suggesting the feasibility of using immunotherapy agents in OTRs with careful conversion of calcineurin inhibitor to mTOR inhibitors and use of prophylactic steroids [122]. In 2011, the CONVERT trial demonstrated reduced incidence of malignancy amongst renal transplant recipients who switched from calcineurin inhibitors to sirolimus-based immunosuppression over two years [123]. A more recent retrospective study by Murray et al. compared the rates of NMSC in almost 5000 kidney and liver transplant patients before and after transitioning to mTOR-based immunosuppression [124]. The authors demonstrated that there was a 50% reduction in NMSCs in both transplant groups, with a median of 3.4 years follow-up in renal transplant recipients [124]. A retrospective review of seven OTRs who were treated with cemiplimab and pembrolizumab demonstrated an overall response rate of 57.1% at median follow-up of seven months [122]. Cemiplimab is of particular significance as it is the only systemic immunotherapy approved for use in both cuSCC and BCC [125]. Despite these promising findings, the high rate of allograft rejection, which ranges from 20–44% in patients treated with checkpoint inhibitors, is concerning [126]. In a systematic review of 57 OTRs treated with checkpoint inhibitors, 21 patients experienced allograft rejection (37%) [127]. PD-1 inhibitors were associated with the highest rates of rejection in this group [127]. Chronically immunosuppressed patients thus continue to be excluded from the majority of randomized clinical trials involving systemic therapies and there remains an urgent unmet need for a safe and effective treatment for these patients.

## 8. Conclusions

The aggressive nature and course of NMSCs in high-risk, immunocompromised patients highlight the need for multidisciplinary management of these patients and measures to prevent and reduce their risk of developing these cancers. Immunomodulation and other targeted therapies have already entered the mainstream of treatments for NMSC in the setting of relapsed and/or metastatic disease. Together, these new therapeutic modalities promise to broaden treatment options for immunocompetent and immunocompromised patients with NMSC and permit a more individualized approach to their treatment.

## Figures and Tables

**Figure 1 cancers-15-03348-f001:**
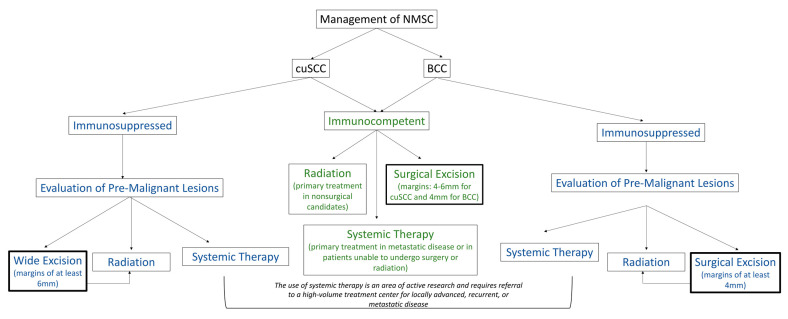
Current strategies for the treatment of cuSCC and BCC. Thicker boxes indicate primary treatment course.

**Table 1 cancers-15-03348-t001:** Clinical Trials of Immunotherapy Agents in Cutaneous Squamous Cell Carcinoma.

NCT Number	Title	Status	Agent(s)	Phases
NCT04620200	Neo-adjuvant Nivolumab or Nivolumab With Ipilimumab in Advanced Cutaneous Squamous Cell Carcinoma Prior to Surgery	Recruiting	Nivolumab and Ipilimumab	Phase 2
NCT04160065	Immunotherapy With IFx-Hu2.0 Vaccine for Advanced Non-melanoma Skin Cancers	Recruiting	IFx-Hu2.0	Phase 1
NCT04632433	Neoadjuvant Plus Adjuvant Treatment With Cemiplimab in Cutaneaous Squamous Cell Carcinoma	Active, not recruiting	Cemiplimab	Phase 2
NCT05574101	A Study of Radiation Therapy and Cemiplimab for People With Skin Cancer	Recruiting	Cemiplimab	Phase 2
NCT05110781	Atezolizumab Before Surgery for the Treatment of Regionally Metastatic Head and Neck Squamous Cell Cancer With an Unknown or Historic Primary Site	Recruiting	Atezolizumab	Phase 2
NCT04329221	Immunotherapy Before Transplantation for Skin Cancer Prevention in Organ Transplant Recipients	Not yet recruiting	Calcipotriol, Vaseline, and Topical 5FU	Phase 2
NCT04204837	Nivolumab for Treatment of Squamous Cell Carcinoma of the Skin	Recruiting	Nivolumab and Nivolumab plus Relatlimab	Phase 2
NCT04428671	Cemiplimab Before and After Surgery for the Treatment of High Risk Cutaneous Squamous Cell Cancer	Recruiting	Cemiplimab	Phase 1
NCT03737721	The UNSCARRed Study: UNresctable Squamous Cell Carcinoma Treated With Avelumab and Radical Radiotherapy	Recruiting	Avelumab	Phase 2
NCT04642287	Immunotherapy After Transplantation for Skin Cancer Prevention in Organ Transplant Recipients	Not yet recruiting	Calcipotriol, Vaseline, and Topical 5FU	Phase 2
NCT03944941	Avelumab With or Without Cetuximab in Treating Patients With Advanced Skin Squamous Cell Cancer	Recruiting	Avelumab and Cetuximab	Phase 2
NCT03565783	Cemiplimab in Treating Patients With Recurrent and Resectable Stage II-IV Head and Neck Cutaneous Squamous Cell Cancer Before Surgery	Recruiting	Cemiplimab	Phase 2
NCT04454489	Quad Shot Radiotherapy in Combination With Immune Checkpoint Inhibition	Recruiting	Pembrolizumab	Phase 2
NCT04163952	Talimogene Laherparepvec and Panitumumab for the Treatment of Locally Advanced or Metastatic Squamous Cell Carcinoma of the Skin	Active, not recruiting	Panitumumab and Talimogene Laherparepvec	Phase 1
NCT05085496	Radiotherapy in Combination With Atezolizumab in Locally Advanced Borderline Resectable or Unresectable Cutaneous SCC	Recruiting	Atezolizumab	Phase 1
NCT04925713	IFx-Hu2.0 for the Treatment of Patients With Skin Cancer	Completed	IFx-Hu2.0	Phase 1
NCT04315701	A PD-1 Checkpoint Inhibitor (Cemiplimab) for High-Risk Localized, Locally Recurrent, or Regionally Advanced Skin Cancer	Recruiting	Cemiplimab	Phase 2
NCT05721755	Combining Radiation Therapy With Immunotherapy for the Treatment of Metastatic Squamous Cell Carcinoma of the Head and Neck	Not yet recruiting	Carboplatin, Cisplatin, Fluorouracil, Paclitaxel, and Pembrolizumab	Phase 3
NCT02978625	Talimogene Laherparepvec and Nivolumab in Treating Patients With Refractory Lymphomas or Advanced or Refractory Non-melanoma Skin Cancers	Recruiting	Nivolumab and Talimogene Laherparepvec	Phase 2
NCT05025813	Neoadjuvant Pembrolizumab in Cutaneous Squamous Cell Carcinoma	Recruiting	Pembrolizumab	Phase 2
NCT05086692	A Beta-only IL-2 ImmunoTherapY (ABILITY) Study	Recruiting	MDNA11 Monotherapy and MDNA11 in Combination with Checkpoint Inhibitor	Phase 1/Phase 2
NCT04576091	Testing the Addition of an Anti-cancer Drug, BAY 1895344, With Radiation Therapy to the Usual Pembrolizumab Treatment for Recurrent Head and Neck Cancer	Recruiting	Elimusertib and Pembrolizumab	Phase 1
NCT05269381	Personalized Neoantigen Peptide-Based Vaccine in Combination With Pembrolizumab for the Treatment of Advanced Solid Tumors, The PNeoVCA Study	Recruiting	Cyclophosphamide, Neoantigen Peptide Vaccine, Pembrolizumab, and Sargramostim	Phase 1
NCT02955290	CIMAvax Vaccine, Nivolumab, and Pembrolizumab in Treating Patients With Advanced Non-small Cell Lung Cancer or Squamous Head and Neck Cancer	Recruiting	Nivolumab, Pembrolizumab, and Recombinant Human EGF-rP64K/Montanide ISA 51 Vaccine	Phase 1/Phase 2
NCT03108131	Cobimetinib and Atezolizumab in Treating Participants With Advanced or Refractory Rare Tumors	Active, not recruiting	Atezolizumab and Cobimetinib	Phase 2
NCT04916002	CMP-001 in Combination With IV PD-1-Blocking Antibody in Subjects With Certain Types of Advanced or Metastatic Cancer	Recruiting	CMP-001 and Cemiplimab-rwlc	Phase 2
NCT04007744	Sonidegib and Pembrolizumab in Treating Patients With Advanced Solid Tumors	Recruiting	Pembrolizumab and Sonidegib	Phase 1
NCT01984892	Treatment of Solid Tumors With Intratumoral Hiltonol (Poly-ICLC)	Terminated	Poly-ICLC	Phase 2
NCT04272034	Safety, Tolerability, Pharmacokinetics, and Pharmacodynamics of INCB099318 in Participants With Advanced Solid Tumors	Recruiting	INCB099318	Phase 1
NCT04242199	Safety, Tolerability, Pharmacokinetics, and Pharmacodynamics of INCB099280 in Participants With Advanced Solid Tumors	Recruiting	INCB099280	Phase 1
NCT03816332	Tacrolimus, Nivolumab, and Ipilimumab in Treating Kidney Transplant Recipients With Selected Unresectable or Metastatic Cancers	Active, not recruiting	Ipilimumab, Nivolumab, Prednisone, and Tacrolimus	Phase 1
NCT04301011	Study of TBio-6517 Given Alone or in Combination With Pembrolizumab in Solid Tumors	Active, not recruiting	TBio-6517 and Pembrolizumab	Phase 1/Phase 2
NCT04596033	TiTAN-1: Safety, Proliferation and Persistence of GEN-011 Autologous Cell Therapy	Terminated	GEN-011, IL-2, Fludarabine, and Cyclophosphamide	Phase 1
NCT05076760	Study of MEM-288 Oncolytic Virus in Solid Tumors Including Non-Small Cell Lung Cancer (NSCLC)	Recruiting	MEM-288 Intratumoral Injection	Phase 1
NCT04799054	A Study of TransCon TLR7/8 Agonist With or Without Pembrolizumab in Patients With Advanced or Metastatic Solid Tumors	Recruiting	TransCon TLR7/8 Agonist and Pembrolizumab	Phase 1/Phase 2
NCT04348916	Study of ONCR-177 Alone and in Combination With PD-1 Blockade in Adult Subjects With Advanced and/or Refractory Cutaneous, Subcutaneous or Metastatic Nodal Solid Tumors or With Liver Metastases of Solid Tumors	Active, not recruiting	ONCR-177 and Pembrolizumab	Phase 1
NCT03633110	Safety, Tolerability, Immunogenicity, and Antitumor Activity of GEN-009 Adjuvanted Vaccine	Completed	GEN-009 Adjuvanted Vaccine, Nivolumab, and Pembrolizumab	Phase 1/Phase 2

**Table 2 cancers-15-03348-t002:** Clinical Trials of Immunotherapy Agents in Cutaneous Basal Cell Carcinoma.

NCT Number	Title	Status	Agent(s)	Phases
NCT04925713	IFx-Hu2.0 for the Treatment of Patients With Skin Cancer	Completed	IFx-Hu2.0	Phase 1
NCT02978625	Talimogene Laherparepvec and Nivolumab in Treating Patients With Refractory Lymphomas or Advanced or Refractory Non-melanoma Skin Cancers	Recruiting	Nivolumab and Talimogene Laherparepvec	Phase 2
NCT05086692	A Beta-only IL-2 ImmunoTherapY (ABILITY) Study	Recruiting	MDNA11 Monotherapy and MDNA11 in Combination with Checkpoint Inhibitor	Phase 1/Phase 2
NCT03816332	Tacrolimus, Nivolumab, and Ipilimumab in Treating Kidney Transplant Recipients With Selected Unresectable or Metastatic Cancers	Active, not recruiting	Ipilimumab, Nivolumab, Prednisone, and Tacrolimus	Phase 1

**Table 3 cancers-15-03348-t003:** Clinical Trials of Systemic and Chemopreventive Agents in Immunocompromised Patients with cuSCC and BCC.

NCT Number	cuSCC or BCC	Agent(s)	Title
NCT03769285	Both	Nicotinamide	Nicotinamide Chemoprevention for Keratinocyte Carcinoma in Solid Organ Transplant Recipients: A Pilot, Placebo-controlled, Randomized Trial
NCT02978625	Both	RP1	An Open-Label, Multicenter, Phase 1B/2 Study of RP1 in Solid Organ and Hematopoietic Cell Transplant Recipients With Advanced Cutaneous Malignancies
NCT02218164	cuSCC	Capecitabine	A Phase 2 Study of Capecitabine or 5-FU With Pegylated Interferon Alpha-2b in Unresectable/Metastatic Cutaneous Squamous Cell Carcinoma
NCT00003611	Both	Acitretin	Chemoprevention Trial of Acitretin Versus Placebo in Solid Organ Transplant Recipients With Multiple Prior Treated Skin Cancers
NCT01358045	BCC	Diclofenac Sodium	Topical Vitamin D3, Diclofenac or a Combination of Both to Treat Basal Cell Carcinoma
NCT00472459	Both	Metvix + PDT	A Multicentre, Randomised Study of Photodynamic Therapy(PDT) With Metvix^®^ 160 mg/g Cream in Immuno-compromised Patients With Non-melanoma Skin Cancer

## Data Availability

Not applicable.

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
