# Peer review of "Management of Cutaneous Head and Neck Squamous and Basal Cell Carcinomas for Immunocompromised Patients"

_cancers, 2023, doi:10.3390/cancers15133348_

Round 1
Reviewer 1 Report (Previous Reviewer 1)
Answers to my requests are satisfactory.
Author Response
Thank you very much for your review of our revised draft.
Reviewer 2 Report (Previous Reviewer 3)
In its revised form, the paper seems interesting to readers.
Fig 1 was not included in this resubmission material.
Author Response
Thank you very much for your feedback. I am attaching figure 1 here and will also ensure that it is included in the final version of the manuscript.

This manuscript is a resubmission of an earlier submission. The following is a list of the peer review reports and author responses from that submission.
Round 1
Reviewer 1 Report
General comments
The topic of the review meanly deals with clinical management of immunosuppressed patients after organ transplantation (e.g., kidney).
Although I think that this review could be very informative for professionals in the field, it seems too long due to with many text redundancies. Also, along the text it is quite difficult to understand which kind of NMSC (i.e., SCC vs BCC) the authors are mentioning.
I am a bit worried with the fact that the authors do not clearly differentiate BCCs vs SCCs along the text. To my knowledge BCCs, are poorly metastatic in the general population so that the distinction between characteristics in these two cuNMSC should more clearly separated. In other words, I would be more positive for a review (meta-analysis) better focused on SCCs.
Detail points, non exhaustive list
- L 38-39: is it respectively?
- L 51-52, what are the rates (%) of transition of AK toward Cu SCCs; “often” is quite vague
- L 57 Of through???
P21 is involved in cell growth arrest which indirectly facilitates e.g. Nucleotide Excision Repair. Furthermore, loss of a master gene of DNA repair, namely XPC, has been frequently observed in cu SCCs (de Feraudy et. al., 2010).
L65-67
Are the authors talking about the Sonic Hedgehog (SHH) pathway? In the majority of BCCs the SHH pathway is altered. Unless I missed it, the GDC-0449 SHH inhibitor is not mentioned.
L 68
The rules of nomenclature of GENES, PROTEINS, mRNAs, should be respected along the manuscript. Same for L 96.
“TIME” should be better explained.
Why PDT is is preferentialy used for treatment of BCCs.
Concerning PD-1 and PDL-1 the text is a bit confusing.
Reviewer 2 Report
The authors reported the results of a review article investigating current knowledge on the pathogenesis of head and neck cutaneous SCC and BCC in immunocompromised patients, evaluating the role of the immune response in disease progression, as well as the therapeutic potential of novel immunotherapies. The manuscript is interesting and well written. However, I have some suggestions.
My comments: - Introduction: the incidence of SCC is increasing as well. - Introduction: the prevalence of BCC and SCC should be reported. - Introduction: generally, immunocompromised patients have an increased risk of several cutaneous diseases and cancers. Please specify. - Introduction: please define what you mean for "immunocompromised patients" - Material and methods: this section is lacking. Please add. - Results: results should be discussed in a separate section. - Discussion: discussion should be improved. The role of hedgehog inhibitors in the management of BCC should be discussed (you should read and cite "10.1007/s12325-022-02044-1" and "10.2340/00015555-3495") - Discussion: treatment guidelines for BCC and SCC should be briefly discussed, highlighting the need for new and effective drugs (10.1016/j.jaad.2017.10.007 and 10.1016/j.ejca.2019.06.003) - Discussion: you should specify that cemiplimab is the only immunotherapy approved both for SCC and BCC (please read and cite 10.1080/14740338.2022.1993819) - Table: tables are adequate - References: references should be improved.
Reviewer 3 Report
The paper addresses head and neck NMSCs (SCC and BCC) in immunocompromised patients.
a) The authors aim to summarize current knowledge regarding the pathogenesis of these NMCSCs.
b) They also examine the potential role of the immune response in disease progression and detail the role of novel immunotherapies in this subset of patients.
The paper needs more focus concerning the author's aims to be more precise and engaging to readers, i.e., with potential translation to routine care of patients:
1) The role of the immune evasion section needs to be better correlated with the rationale for immunotherapy agents in cuSCC (Table 1) and BCC (Table 2);
2) Systemic and chemopreventive agents in immunocompromised patients with cuSCC and BCC (Table 3) must also be correlated with the pathogenesis section.
3) Display schematically significant management procedure differences between NMSCs in immunocompetent and immunocompromised patients (e.g., OTRs).
